Bird protection treatments reduce bird-window collision risk at low-rise buildings within a Pacific coastal protected area

http://orcid.org/0000-0002-7207-838X De Groot Krista L. 1 krista.degroot@ec.gc.ca
Wilson Amy G. 2
McKibbin René 1
Hudson Sarah A. 1
Dohms Kimberly M. 1
Norris Andrea R. 1
Huang Andrew C. 1
Whitehorne Ivy B. J. 1
Fort Kevin T. 1
http://orcid.org/0000-0002-5599-6234 Roy Christian 3
Bourque Julie 3
Wilson Scott 1 2
1 Pacific Wildlife Research Centre, Environment and Climate Change Canada , Delta, British Columbia , Canada
2 Department of Forest and Conservation Sciences, University of British Columbia , Vancouver, British Columbia , Canada
3 Environment and Climate Change Canada , Gatineau, Quebec , Canada
Kramer Donald
Electronic publication date: 2022 Mar 22
Publication date: 2022
Volume: 10
Electronic Location ID: e13142
Received 2021 Dec 7; Accepted 2022 Feb 28
Copyright: © 2022 De Groot et al.
Copyright year: 2022
Copyright holder: De Groot et al.
License: This is an open access article distributed under the terms of the Creative Commons Attribution License, which permits unrestricted use, distribution, reproduction and adaptation in any medium and for any purpose provided that it is properly attributed. For attribution, the original author(s), title, publication source (PeerJ) and either DOI or URL of the article must be cited.
License URL: https://creativecommons.org/licenses/by/4.0/

Keywords: Bird-window collisions, Mitigation, Effectiveness, Feather Friendly, Real-world testing, Year-round monitoring, ORNILUX, UV, Retrofits, Avian collisions

Funding: Environment and Climate Change Canada (ECCC) Funding for research, Feather Friendly® materials, and installation provided by Environment and Climate Change Canada (ECCC). The funders had no role in study design, data collection and analysis, decision to publish, or preparation of the manuscript.

==============================
Background

In North America, up to one billion birds are estimated to die annually due to collisions with glass. The transparent and reflective properties of glass present the illusion of a clear flight passage or continuous habitat. Approaches to reducing collision risk involve installing visual cues on glass that enable birds to perceive glass as a solid hazard at a sufficient distance to avoid it.

Methods

We monitored for bird-window collisions between 2013 and 2018 to measure response to bird protection window treatments at two low-rise buildings at the Alaksen National Wildlife Area in Delta, British Columbia, Canada. After 2 years of collision monitoring in an untreated state, we retrofitted one building with Feather Friendly® circular adhesive markers applied in a grid pattern across all windows, enabling a field-based assessment of the relative reduction in collisions in the 2 years of monitoring following treatment. An adjacent building that had been constructed with a bird protective UV-treated glass called ORNILUX® Mikado, was monitored throughout the two study periods. Carcass persistence trials were conducted to evaluate the likelihood that carcasses were missed due to carcass removal between scheduled searches.

Results and Conclusions

After accounting for differences in area of glass between the two buildings, year, and observer effects, our best-fit model for explaining collision risk included the building’s treatment group, when compared to models that included building and season only. We found that the Feather Friendly® markers reduced collision risk at the retrofitted building by 95%. Collision incidence was also lower at the two monitored façades of the building with ORNILUX® glass compared to the building with untreated glass. Although more research is needed on the effectiveness of bird-protection products across a range of conditions, our results highlight the benefit of these products for reducing avian mortality due to collisions with glass.

Introduction

Each year in North America, billions of birds suffer direct anthropogenic mortality, of which collisions with glass on buildings are a major source (Loss, Will & Marra, 2015). Building collisions are responsible for an estimated 365–988 million avian deaths per year in the United States and 16–42 million deaths per year in Canada (Machtans, Wedeles & Bayne, 2013; Loss et al., 2014). As the built environment continues to grow worldwide (Seto, Güneralp & Hutyra, 2012), research effort has also been steadily increasing to identify the temporal, spatial, and taxonomic correlates of collision risk. Mortalities from collisions are often higher during the migratory periods compared to the stationary breeding and the overwintering stages of the annual cycle (Borden et al., 2010; Machtans, Wedeles & Bayne, 2013; Loss et al., 2014, 2019; Kummer & Bayne, 2015). Therefore, even populations of species that spend the majority of their lifecycle in areas that lack anthropogenic structures, may suffer collision mortality losses during migratory stopovers in urban and rural areas (Pennington, Hansel & Blair, 2008; Riding, O’Connell & Loss, 2020). Collision mortality can also be substantial throughout the nonbreeding period in developed regions that support high densities of overwintering birds (De Groot et al., 2021).

Some species suffer disproportionate levels of collision-related mortality relative to their local abundance, indicating that species-specific traits increase the susceptibility of specific taxa (Loss et al., 2014; Wittig et al., 2017; Elmore et al., 2020; De Groot et al., 2021). For example, nocturnal migrants (Arnold & Zink, 2011; Loss et al., 2014) and forest-dwelling insectivorous songbirds appear to have increased vulnerability to collisions (Wittig et al., 2017; Elmore et al., 2020). Omnivorous birds, particularly those that switch to a diet of fruit outside of the breeding season, have also been shown to be more susceptible to collisions during the nonbreeding period (Brown et al., 2019; De Groot et al., 2021). Regional-levels of collision mortality can be influenced by geographic features such as large lakes, building densities (Machtans, Wedeles & Bayne, 2013; Hager et al., 2017), and the lightscape (Van Doren et al., 2017; Horton et al., 2019). These regional-scale factors interact with local-scale factors such as neighbourhood and building-level collision risk (Hager et al., 2013; Riding, O’Connell & Loss, 2020). Increased collision risk occurs when buildings have features that attract birds to buildings, such as surrounding vegetation (Klem, 1989; Hager et al., 2013; Riding, O’Connell & Loss, 2020), feeders (Dunn, 1993; Kummer, Bayne & Machtans, 2016), fruit trees (Brown, Hunter & Santos, 2020), and artificial light at night (Van Doren et al., 2017; Horton et al., 2019). Once birds are in close proximity to buildings, other factors such as architectural features (e.g. glass corners, alcoves and corridors), and glass façade characteristics (length, height) further influence collision risk (Hager et al., 2013; Klem et al., 2009; Ocampo-Peñuela et al., 2016; Riding, O’Connell & Loss, 2020). Resident birds may learn where these hazards are located (Sheppard, 2019), but this prior knowledge could be lacking for migrants and many overwintering birds that may move across large areas in search of ephemeral resources.

Collisions are thought to occur because reflections of sky or vegetation and the transparent nature of glass gives birds the illusion of an unobstructed route (Klem, 1989; Sheppard, 2019). Strategies for collision reduction, therefore aim to provide sufficient visual cues to allow birds to perceive a solid barrier and avoid the hazard (Loss et al., 2019). The use of architectural features such as grilles, solar shading, architectural mesh, external screens and ceramic fritting provide cost-effective solutions that may reduce collision risk (Canadian Standards Association, 2019; American Bird Conservancy, 2019), and also align with other sustainability and occupant objectives, e.g., enhancing energy conservation and occupant privacy. For existing buildings, mitigation approaches involve external window surface treatments such as vertical parachute cords, screens, adhesive opaque and UV films, ceramic fritting or other visual markers (Klem & Saenger, 2013; Rössler, Nemeth & Bruckner, 2015; Sheppard, 2019; Swaddle et al., 2020). Visual markers should cover the entire surface of glass with 5 cm or less between markers, to discourage birds from attempting to navigate around or between the perceived barriers (Klem, 2009; Sheppard, 2019). Frontal vision in most birds is low resolution, and therefore, the color of visual markers must provide as much contrast as possible, in order to be conspicuous to birds (Martin, 2011).

One potential barrier to the uptake of these surface treatments of glass by the general public, is the perception that they may interrupt views for human occupants within buildings. An alternative is to use glass or window treatments with an integrated ultraviolet (UV) signal. The UV treatment reflects at a spectrum in the 300–400 nm range which is visible to most bird species, but not humans (Klem, 2006; Martin, 2011), thereby providing a relatively unobstructed view for building occupants (Klem & Saenger, 2013). Although these products attempt to integrate anecdotal information, collective experience, and existing knowledge on avian vision and obstacle avoidance (Martin, 2011), there has been little formal evaluation of the effectiveness of different solutions across a range of conditions, buildings and geographical areas. Standardized flight-tunnel protocols, which allow birds to fly towards a clear control panel or a panel treated with markers in a range of sizes, colours and configurations, have been developed to allow for large numbers of birds to be tested under controlled conditions (Sheppard, 2019). However, because flight tunnel-based trials cannot capture exact field conditions, they may not be predictive of field effectiveness, potentially leading to an overestimation of their efficacy (Klem & Saenger, 2013; Swaddle et al., 2020). Products assessed across a range of seasons and under a range of natural lighting and field conditions may therefore increase the reliability of product evaluations (Klem & Saenger, 2013; Rössler, Nemeth & Bruckner, 2015; Swaddle et al., 2020). Furthermore, bird-collision deterrent products are best evaluated if collision data are collected both before and after the application of bird protection products (Sheppard, 2019; Swaddle et al., 2020). To our knowledge, only one study to date has applied such an experimental design to evaluate the reduction in collisions after the application of external adhesive markers (described in Brown, Hunter & Santos, 2020; Brown, Santos & Ocampo-Peñuela, 2021).

In this study, we present in situ performance data on collision risk from our study site in Delta, British Columbia, Canada across all seasons between 2013 and 2018 for two commercially-available bird protection products: Feather Friendly® (FF) external adhesive markers (Feather Friendly Technologies Inc., Mississauga, Ontario, Canada) and ORNILUX® Mikado N33 UV patterned bird-protection glass (Arnold Glas, Remshalden, Germany). Both ORNILUX® and FF meet the American Bird Conservancy’s criterion for bird-friendly glass (American Bird Conservancy, 2021), based on tunnel testing. Our objective was to evaluate the effectiveness of bird protection products under natural conditions. We first examined whether retrofitting a building with FF treatment reduced collision risk by comparing collision rates at all 11 façades on a single building for 2 years before, and 2 years after applying the FF external adhesive markers to all glass surfaces of the building. Our secondary objective was to assess the collision rate over the same time period at an adjacent building constructed using ORNILUX® glass.

Materials and Methods

Study area

We conducted collision monitoring at two buildings of the Pacific Wildlife Research Centre (PWRC), referred to as the Science Complex and the Annex (Fig. 1). The buildings are situated within the Alaksen National Wildlife Area (hereafter Alaksen) (49.098, −123.179°) located on Westham Island within the Fraser River Delta in Delta, British Columbia, Canada. Alaksen spans 349 ha on Westham Island and overlaps with a portion of the 300 ha George C. Reifel Migratory Bird Sanctuary. The habitat at Alaksen includes estuarine and freshwater marsh habitats, a mixture of deciduous and coniferous trees and shrubs interspersed along dikes, and 140 ha of agricultural fields (Environment and Climate Change Canada, 2020). Alaksen is an important migratory stopover and overwintering site for birds, receiving designation as a Ramsar Wetland of International Significance and Important Bird Area, with over 240 bird species documented to occur, including eight avian federal Species at Risk (Environment and Climate Change Canada, 2020).

Figure 1 Location of buildings at the Pacific Wildlife Research Centre within the Alaksen Wildlife Management Area, Delta, British Columbia, Canada.

(A) The northwestern facing façades 2 and 3 were monitored at the Annex building. (B) All façades (1–11) were monitored for collisions at the Science Complex. Data source: City of Delta and the Canadian Wildlife Service, Environment and Climate Change Canada.

The Science Complex is a two-story building with a 650 m2 footprint (Figs. 1 and 2). The building contains 11 façades, 202 glass window panels, and seven doors with window inserts, for a total glass area of 110.5 m2. In 2016, following the first 730 day collision monitoring study period, all external window panels were treated with Feather Friendly® 1 cm diameter white circular adhesive markers. Feather Friendly® (hereafter FF; Feather Friendly Technologies Inc., Mississauga, Ontario, Canada) is a durable non-film, chromatic adhesive marker that is designed to be applied to the exterior surface of glass by homeowners, or by professional installers for large surfaces of commercial buildings. Markers were applied in a grid pattern with 5 cm spacing from marker centre to centre, across the entire exterior surface of all windows (Fig. 2B). The Annex is a two-story building with a 340 m2 footprint, built in 2013. All external windows and glass doors of the Annex were constructed using ORNILUX® Mikado N33 Bird Protection Glass (hereafter ORNILUX®; Arnold Glas, Remshalden, Germany); a manufactured glass product with a UV signal applied to an interior surface of insulated glass units (surface 2; Arnold Glas, personal communication, 2021). At the Annex, we chose to monitor the two façades that did not have metal grilles covering a portion of the windows and that best matched the Science Complex in terms of ground vegetation and substrate within 2 m of the building façade (Fig. 3). Monitored façades at the Annex (façades 2 and 3; Figs. 1 and 3) were located on the northwest aspect of the building and contained 37 window panels, including five glass doors with a total glass area of 35.1 m2. The Science Complex and the Annex buildings are less than 15 m apart and have comparable surrounding vegetation structure and proximity to water bodies at the landscape scale (Fig. 1).

Figure 2 The Science Complex at the Alaksen National Wildlife Area, Delta, British Columbia.

(A) A total of 31% of collisions occurred during the pre-treatment (2013–2015) time period at façade 8 (see also Fig. 1). (B) An example of Feather Friendly® 1 cm diameter circular window markers. Photo credit: K. De Groot.

Figure 3 Monitored façades of the Annex building with Ornilux® glass at the Alaksen National Wildlife Area, Delta, British Columbia, Canada.

Photo credit: K. De Groot.

Survey protocol

Surveys were conducted by Environment and Climate Change Canada staff and affiliated students. For each survey, a participant walked all façades one to eleven of the Science Complex and façades two and three of the Annex (Fig. 1), searching for stunned birds, intact carcasses, partially scavenged carcasses, or feather piles within 2 m of building façades, and feather smears on first story windows (Hager & Cosentino, 2014). Feather piles were defined as a minimum of 10 feathers found within a 0.5 m diameter area (Ponce et al., 2010). If feather smears were present in addition to a carcass or feather pile at the same façade, these were treated as a single collision. Surveys were conducted in the afternoon to maximize detection, because most collision mortalities are expected to occur in the early morning (Hager & Cosentino, 2014). Participants recorded the time and date of the survey, type of evidence located (carcass, feather smear, feather pile or stunned birds) and species identification if possible. All evidence, including feather smears on first story windows, was then removed by collision monitors to avoid double counting in subsequent surveys. Incidental reports of bird collisions by staff working at either building were recorded separately from standardized surveys and were not included in statistical analyses.

A total of 726 surveys were conducted over 4 years; 2–5 times per week at both the Science Complex and the Annex for 2 years (730 days) before the FF treatment of the Science Complex (pre-treatment period: April 3rd, 2013 to April 2nd, 2015; n = 362 surveys), and 2–5 times per week at both buildings for 2 years (730 days) following FF treatment (post-treatment period: September 26th, 2016 to September 25th, 2018; n = 364 surveys). Surveys were conducted throughout the year with 180 in the spring (March 21–June 20), 205 in the summer (June 21–September 21), 169 in the fall (September 22–December 20), and 172 in the winter (December 21–March 21) (Table S1). A clean-up survey was conducted the day before the start of each of the two survey periods (i.e., on April 2nd, 2013 and September 25th, 2016) to remove any collision evidence that had accumulated prior to the survey periods.

Scavenging of carcasses can negatively bias collision monitoring results if carcasses are removed prior to scheduled surveys (Klem et al., 2004; Riding & Loss, 2018). Carcass persistence was measured to evaluate whether carcasses were being missed, by placing thawed carcasses of previously window-killed birds at randomly selected locations along building façades. The persistence of carcasses was monitored twice daily for the first 3 days and then during scheduled surveys (2–5 days per week) on subsequent days until carcasses were not detectable, or after the trial had run for 11 days, whichever came first.

Statistical analysis

All data manipulations were performed using the tidyverse package (Wickham et al., 2019). We used generalized linear mixed models, incorporating only the standardized survey collision monitoring data, assuming a Poisson error distribution with a log-link function using the lme4 package (Bates et al., 2015) in R version 4.0.1 (R Core Team, 2019). We used the bobyqa iterative optimizer in lme4, which facilitates the convergence of GLMM models with bounded variables using an iterative quadratic approximation (Powell, 2009; Bates et al., 2015). Although each survey day included all façades at the Science Complex and the two monitored façades at the Annex, data were pooled across façades and were treated at the building level. Data were compiled into four groups: Pre-FF treatment (2013–2015) and post-FF treatment (2016–2018) period for the Science Complex and over the same time blocks of 2013–2015 and 2016–2018 for the Annex building with ORNILUX® glass (Table S2). The rationale of this grouping was to ensure equal partitioning of inter-annual variance between the two buildings that could influence our assessment of collision rate such as weather, bird population fluctuations and behavior. These treatment groups were modelled as a fixed effect, as was season. The influence of year and observer were accounted for by modelling them as random effects. An offset correction was used to account for the differences in glass area between the Science Complex (FF) and the two surveyed façades of the Annex (ORNILUX® glass).

We evaluated support for three models as shown in Table 1. We compared a baseline model, which contained only year and observer random effects, against three alternatives that added fixed effects. These additions included tests of building and season, season only, and the full model with treatment effects (conventional glass: Science Complex pre-treatment period; FF: Science Complex post-treatment period; or ORNILUX® glass: Annex building both periods). Model support was based on Akaike’s Information Criterion (AIC), where models with ∆AIC < 2 were considered equally plausible (Burnham & Anderson, 2002). Incidence rate-ratios (IRR) were then calculated using sjplot (Lüdecke, 2020) and reflect the relative increase (IRR > 1) or decrease (IRR < 1) in the incidence of collisions within a categorical group, relative to a reference group, which we selected as the untreated glass at the Science Complex.

Table 1 Akaike’s Information Criterion (AIC) model selection results for evaluating factors influencing the number of avian collisions detected at two buildings, corrected for differences in the area of glass between the two buildings.

	Model structure	AIC	∆AIC	w	df	
A	treat.grp + season + (1|year) + (1|obs)	335.95	0	1.00	9	
B	season + (1|year) + (1|obs)	349.63	13.68	0	6	
C	building + season + (1|year) + (1|obs)	350.25	14.30	0	7	
D	(1|year) + (1|obs)	352.78	16.83	0	3	
Note:

Fixed effects: treat.grp = building by treatment period: Science Complex conventional glass pre-treatment (2013–2015), Science Complex FF post-treatment (2016–2018), Annex ORNILUX® (2013–2015), Annex ORNILUX® (2016–2018); building = Science Complex or Annex; season = Winter, Fall, Spring or Summer. Random effects: 1|year = monitoring year, 1|obs = observer. Offset = window area ∆AIC = change in AIC relative to top model, w = Akaike weight, and df = degrees freedom.

Results

We recorded a total of 71 collisions involving 17 species, over the 4 years of our study. Forty-two of these collisions were known mortalities (carcasses and feather piles) and 27 were collisions for which fate was unknown (19 feather smears and eight stunned birds; Table S2). We found that collisions were most frequent in winter and fall, despite fewer surveys being conducted during these seasons, compared to spring and summer (Tables S1 and S3). Collisions were also not distributed equally across façades. Before FF application on the Science Complex windows, 62% of collisions detected during standardized surveys occurred at façades five (17%), eight (31%) and nine (14%) (Figs. 1 and 2A).

We found that carcass persistence was similar across the two time periods (2013–2015 vs 2016–2018) and across the two study buildings. Carcasses persisted for a mean of 8.2 days (SE = 1.2; range 2–11 days) during the 2013–2015 study period and for a mean of 7.6 days (SE = 1.2 days; range 1–11) during the 2016–2018 study period. Mean carcass persistence was 7.6 days (SE = 1.0; range 1–11 days) and 8.5 days (SE = 2.2 days; range 1.3–11 days) at the Science Complex and the Annex, respectively.

Competing models, based on standardized survey data corrected for differences in area of glass, compared simpler models to the full model. The full model, testing the combination of the building’s treatment group (conventional glass, FF or ORNILUX® glass over both time periods) and season, demonstrated the best model fit (Row A, Table 1) and outcompeted the model that included the random effects, year and observer only (Row D, Table 1), the model adding season (Row B), and a model adding building and season (Row C). We found that the Incidence Rate Ratio (IRR) for collisions was significantly lower at the Science Complex in the 2 year period following FF treatment, representing a 95% reduction in collision risk in 2016–2018 (IRR = 0.05; Fig. 4) compared to the 2 year pre-treatment period (2013–2015) when only conventional glass was present. Collisions at the Annex ORNILUX® glass façades in 2013–2015 were significantly less likely to occur compared to untreated conventional glass at the Science Complex during the same time period (71% reduction in collision risk, IRR = 0.29, Fig. 4). However, the confidence intervals for IRR overlapped with the conventional glass reference, indicating that the reduction in collision risk at the Annex ORNILUX® glass façades in the 2016–2018 period was not statistically significant when compared to untreated conventional glass (66% reduction in collision risk, IRR = 0.34, Fig. 4).

Figure 4 Incident Rate Ratios (IRR) of collision risk derived from best-fit model data corrected for glass area.

IRRs are compared to conventional glass at the Science Complex 2013–2015 (IRR = 1; solid vertical line) and reflect the increase (IRR > 1) or decrease (IRR < 1) in the risk of collisions within a categorical group. IRR values furthest from 1 indicate the greatest reduction in collision risk compared to untreated conventional glass.

Discussion

Our study provides evidence of the effectiveness of two bird protection products utilizing opaque, adhesive markers and UV-treated glass, for the reduction of bird collisions under natural field conditions. The application of Feather Friendly® (FF) adhesive circular markers to conventional glass windows at the Science Complex led to a 95% reduction in collision risk. Our results are consistent with other studies showing significant reductions in collisions following the application of FF markers (FLAP Canada, 2018; Winton, Ocampo-Peñuela & Cagle, 2018; Brown et al., 2019; Brown, Santos & Ocampo-Peñuela, 2021). The Annex windows were constructed with ORNILUX® glass, so the absence of pre-treatment collision data prevented a within-building comparison of change in collision risk. However, the product shows promise in that, relative to conventional glass at an adjacent building, ORNILUX® glass also demonstrated a reduced collision incidence risk.

The effectiveness of Feather Friendly® markers is due to the disruption of the reflectivity of the external surface; providing a visual signal to birds that an impassable barrier is present. To discourage birds from attempting to navigate between the perceived barriers, the visual markers must be applied across the entire surface of glass and inter-marker spacing should be less than 5 cm from centre to centre (Klem, 2009; Sheppard, 2019). These recommendations were followed in the FF application at the Science Complex. The UV patterns on the ORNILUX® Mikado N33 glass also adhere to the spacing recommendations, but the UV patterns in the N33 product are applied to an interior surface of insulated glass units. Patterns placed on interior window surfaces may be less visible under conditions of high reflectance (Sheppard, 2019), or if the outside light exceeds the building’s interior lighting (Klem & Saenger, 2013). An in situ comparison of ORNILUX® and fritted glass also suggested that both treatments reduced collisions compared to untreated windows at other buildings, but fritted glass had fewer collisions than both ORNILUX® glass and untreated glass (Brown, Hunter & Santos, 2020).

Compared to opaque FF markers, windows constructed with UV patterns offer relatively transparent views for humans. However, UV-based products have several limitations that warrant consideration. First, for UV-based signals to be detected by birds, there must be at least 20–40% reflectance within the 300–400 nm range (Klem & Saenger, 2013). UV light levels vary throughout the day, season, and latitude, and UV light levels may be low during the highest risk time for collisions, such as the morning (Loss et al., 2019; Sheppard, 2019). Second, the effectiveness of the UV-signal depends on birds being UV-sensitive. Passerines are UV-sensitive, while other taxa such hummingbirds have lower sensitivity (Ödeen & Håstad, 2013), but also suffer high collision mortalities (Schneider et al., 2018; Loss et al., 2019). Therefore, reducing collision mortality in species with a reduced UV sensitivity would require alternative mitigation approaches. Bird protection products such as Feather Friendly® and UV glass products such as ORNILUX® come in a range of configurations, densities of markers and location of UV signal within insulated glass units, all of which may confer different levels of protection against bird collisions. Effectiveness of products may be further influenced by building and landscaping characteristics, season, latitude, weather, and differences in bird communities. Therefore, additional field studies across a range of sites and conditions would be beneficial (Sheppard, 2019; Elmore et al., 2020; Riding, O’Connell & Loss, 2020; Brown, Santos & Ocampo-Peñuela, 2021). The consolidated ownership of buildings, building ownership stability and motivated occupants make government facilities, colleges and universities ideal locations for multi-year field evaluations of bird protection products. Furthermore, the availability of systematically collected, or incidental collision data would be exponentially increased with the involvement of community scientists (Loss et al., 2015) using existing platforms such as the Global Bird Collision Mapper (FLAP Canada, 2021b), iNaturalist (Winton, Ocampo-Peñuela & Cagle, 2018; California Academy of Sciences & National Geographic, 2021), and dBird (New York Audubon, 2021).

Our first caveat is that we lacked the ability to conduct a within-building, pre- and post-treatment comparison of ORNILUX® glass. The Science Complex and Annex are structurally different buildings in features that are known to influence collisions, such as proportion of glass (Klem, 2009; Riding, O’Connell & Loss, 2020). Although we included an offset correction for the differences in glass area to compare the ORNILUX® to untreated conventional glass at the Science Complex, we lacked the statistical power to account for the potential influence of a range of additional variables. Therefore, the comparison of collision risk of ORNILUX® glass relative to our adjacent building with untreated conventional glass should be interpreted with caution.

Secondly, as we have limited carcass persistence data, we were not able to calculate mortality estimates adjusted for biases due to scavenging of carcasses. However, our available carcass persistence data suggests that carcass removal did not vary between the 2013–2015 and 2016–2018 study periods, or between the Science Complex and Annex buildings. In addition, the average carcass presence of 7.6–8.5 days in our carcass persistence trials indicates that removal would be unlikely to affect detection, because our surveys occurred more frequently than the average time that it took for carcasses to disappear at our survey site. Hager, Cosentino & McKay (2012) reported carcass persistence ranges of 11.8 ± 7.2 days in Illinois during non-winter seasons, which is comparable to our results. However, persistence times were much shorter at a nearby study in Vancouver, British Columbia, where the shortest median durations were markedly different (0.81 days in fall; De Groot et al., 2021). These differences highlight the importance of collecting site-specific carcass persistence data, to evaluate potential biases in mortality data derived from carcass surveys.

Determining the effectiveness of bird-protection products is crucial, but these efforts should also be accompanied by incentives to adopt these mitigations. Bird-friendly building design standards are mandatory in an increasing number of North American municipalities such as Toronto, Ontario and New York City, New York (American Bird Conservancy, 2022). These legislative standards, in addition to a 2013 judgement (Liat Podolsky “Ecojustice” v. Cadillac Fairview) that accidental deaths of federally threatened and endangered species due to collisions with buildings constitutes an offence under Section 32(1) of Canada’s Species At Risk Act (2013), fueled an increased demand for collision mitigation products in Canada. Compliance can be further encouraged by emphasizing multiple benefits of certain collision mitigation solutions such as reduced carbon footprint, and reduction of building operational costs. For example, retrofits of ceramic frit can reduce collision mortality by 90%, but also offer substantial energy savings and retained aesthetics for a proportionately small financial investment (Piselli, 2020). Furthermore, substantial reductions in mortality and savings in energy use can be accomplished by reducing night lighting, as well as focusing funding resources on mitigating the portions of buildings that pose a disproportionately high collision risk (Loss et al., 2019; FLAP Canada, 2021a).

The buildings in this study are comparable to rural residences and low-rises, both of which are cumulatively associated with high collision mortality (Bayne, Scobie & Rawson-Clark, 2012; Machtans, Wedeles & Bayne, 2013; Loss et al., 2014). Reducing mortalities at residential and low-rise buildings could be achieved through increased public awareness of both the scale of window mortality, and the range of simple and inexpensive solutions available to address this mortality, including hanging cords in front of windows (Acopian Bird Savers), or applying designs on windows with tempera paint or oil-based markers (American Bird Conservancy, 2021). The Feather Friendly® markers used in this study are available in ‘Do-It-Yourself’ kits that are easily applied, removable, and cost less than $1 USD per square foot of window. Therefore, members of the public could treat problematic windows at a residence or business for less than $40 USD, which is the approximate amount that moderately engaged respondents were willing to pay to prevent collisions at their homes (Warren, 2013). Promoting these products to the public could be accomplished through partnerships with the bird-feeding industry, home improvement retailers and rebates similar to those offered for home energy-efficiency products.

Conclusions

Feather Friendly® markers reduced collision risk by 95% in a within-building comparison before and after treatment, while a between-building comparison of ORNILUX® glass to conventional untreated glass suggest that a 66–71% reduction in collision occurrence could be possible with this product. Future research focusing on the natural conditions that alter the field performance of mitigation products is crucial for product improvement and ensuring that choice of mitigation product is the best overall match to local conditions, priorities and constraints. Generating the large amounts of data needed to address these knowledge gaps will require interdisciplinary collaboration and community scientist involvement. Finally, prioritizing research towards mitigation efforts that are inexpensive, simple, and/or synergistic with other priorities such as energy conservation, or that engage communities, such as art applied to building glass, are likely to have the greatest cumulative conservation impact.

Supplemental Information

Supplemental Information 1 Summary of collision survey effort between two 730-day periods (2013–2015 and 2016–2018).

Each survey included all 11 façades of the Science Complex and 2 façades of the Annex, at the Pacific Wildlife Research Centre, Delta, BC, Canada.

Click here for additional data file.

Supplemental Information 2 Collision detections and species affected at the Science Complex (untreated conventional glass - April 3rd, 2013 to April 2nd, 2015 and Feather Friendly® treated glass - September 26th, 2016 and September 25th, 2018) and the Annex (Ornilux[sup]®.

Click here for additional data file.

Supplemental Information 3 Summary of collisions detected during standardized surveys across seasons 2013–2015 and 2016–2018 for the Science Complex and Annex.

Click here for additional data file.

Supplemental Information 4 Bird protection treatments reduce collision risk.

Click here for additional data file.

We thank Natalia Ocampo-Peñuela, Donald Kramer and an anonymous reviewer for comments that substantially improved this manuscript. We thank Pat Bishop, Saul Schneider and Nikolas Fehr, for supporting the installation of bird-friendly glass and mitigation products at the Pacific Wildlife Research Centre, and Courtney Albert for provision of student assistants across multiple summers. We are grateful to the many additional Environment and Climate Change Canada staff and students that assisted the authors with collision monitoring, as well as Peter Jacques for substantial support in creating and editing figures. This study was conducted on the traditional and unceded territories of the Musqueam, Stz’uminus, Stó:lō, Quw’utsun, and Tsawwassen Peoples, and the Hul’qumi’num Treaty Group.

Additional Information and Declarations

Competing Interests

Author Contributions

Data Availability

The authors declare that they have no competing interests.

Krista L. De Groot conceived and designed the experiments, performed the experiments, prepared figures and/or tables, authored or reviewed drafts of the paper, and approved the final draft.

Amy G. Wilson analyzed the data, prepared figures and/or tables, authored or reviewed drafts of the paper, and approved the final draft.

René McKibbin performed the experiments, prepared figures and/or tables, authored or reviewed drafts of the paper, and approved the final draft.

Sarah A. Hudson performed the experiments, authored or reviewed drafts of the paper, and approved the final draft.

Kimberly M. Dohms performed the experiments, authored or reviewed drafts of the paper, and approved the final draft.

Andrea R. Norris performed the experiments, authored or reviewed drafts of the paper, and approved the final draft.

Andrew C. Huang performed the experiments, authored or reviewed drafts of the paper, and approved the final draft.

Ivy B. J. Whitehorne performed the experiments, authored or reviewed drafts of the paper, and approved the final draft.

Kevin T. Fort performed the experiments, authored or reviewed drafts of the paper, and approved the final draft.

Christian Roy analyzed the data, authored or reviewed drafts of the paper, and approved the final draft.

Julie Bourque analyzed the data, authored or reviewed drafts of the paper, and approved the final draft.

Scott Wilson analyzed the data, authored or reviewed drafts of the paper, and approved the final draft.

The following information was supplied regarding data availability:

The collision data by building, treatment group and monitoring period are available in the Supplemental File.

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
