# Peer review of "Bird protection treatments reduce bird-window collision risk at low-rise buildings within a Pacific coastal protected area"

_PeerJ, doi:10.7717/peerj.13142_

## Round 0.1 · original submission · Major Revisions

This study compared the number of bird collisions with windows for two years before and two years after the windows were treated with one collision-reducing method (adhesive markers). In addition, a comparison was made with a nearby building treated with an alternative method (UV reflectance) over the same two two-year periods. It contributes to a relatively small number of studies of the effectiveness of bird-friendly windows under real world conditions.

Both reviewers found the study to be a strong and well-presented study. Indeed, it is one of the best-presented manuscripts I have encountered in over 140 that I have edited for PeerJ. In general, it is clear and concise, and I have only a few suggestions for improvements. However, both reviewers also suggest some additional analyses that could clarify the interpretation.

You may treat my comments below as if they were a third review, i.e., make changes if appropriate or explain why changes are not needed.

Editor’s Comments
Title: I am not sure that the name of the institution adds anything to the title Can you replace the name by a relevant characteristic (e.g., ‘low rise buildings near a bird sanctuary’)?
Abstract. You don’t mention the study of carcass persistence in the Abstract. Is this important enough for other researchers that it should be mentioned (even though it could be quite location-specific)?
L43. Add a new subhead (Conclusions) for the final sentence of the Abstract or change the Results subhead to ‘Results and Conclusions’.
L55. comma before ‘therefore’
L235. Start the sentence with ‘In the two years before FF treatment, . . .’ so that it is parallel with the next sentence and emphasizes the before-after comparison.
Discussion: Unlike Reviewer 2, I was not struck by problems with the organization of the Discussion. You can check your topic outline for coherence and perhaps ask a colleague or two what they think of the suggestion and then decide whether change is needed.
Discussion: You don’t mention carcass persistence in the Discussion except as a caveat that you don’t have enough data to correct for it. I wonder if a bit more discussion is worthwhile. Have others undertaken such measures? If so, how similar are the values? If not, should they be included in studies where possible? Is there anything to say about the causes of carcass loss?
L299. Run-on sentence. Start a new sentence with ‘However,’
L334. (same issue as L299)
Conclusion: Like Reviewer 2, I agree that this section is too long. One paragraph is more appropriate. A single, more concise (perhaps half the length), paragraph incorporating the current first and last paragraphs would be appropriate. The second paragraph of the Conclusions seems appropriate for a final paragraph of the Discussion.
References: A few article titles have capital letters throughout; please check.
Fig. 1. Not clear what the text is in the inset box. If it is a photo credit, could it go in the caption? Should there be a credit for the main photo of the buildings?
Fig. 2. There are more than two facades in Panel A. Could you add numbers or refer to figure 1 in the caption? Photo credit? The last sentence is not needed; it overlaps Methods.
Fig. 3. I suggest: ‘Monitored facades of the Annex building with ORNILUX® windows at the Alaksen National Wildlife Area, Delta, British Columbia, Canada.’ The last sentence is not needed; it overlaps Methods.
Fig. 4. I prefer keeping the figure rather than replacing it with a table as suggested by a reviewer, but Reviewer 1 has a suggestion that might incorporate the suggestion of Reviewer 2 and the information conveyed by this figure. L4 add: ‘black vertical line’. As the reviewer points out, using only color as a reference means that black and white copies will be ambiguous and that color-blind readers may not be able to follow. I suggest adding descriptions of positions (top, middle, bottom). If you keep the figure, I think that a more visually attractive figure is possible. The three lines do not need to be spaced so far apart. The labels on the y-axis could go above the lines instead of out to the left. The fonts of axis labels on x- and y-axes and the tick marks should be larger to be more easily read on a photocopy. The background doesn’t need to be gray, and the vertical and horizontal lines within the panel are not needed.
Table 1. Vertical lines do not ordinarily appear in tables and do not seem necessary here. Table headings can be underlined. Lines are not needed between the 4 models.

Reviewer 1 ·

Excellent Review

This review has been rated excellent by staff (in the top 15% of reviews)
EDITOR COMMENT
This was an excellent review, strongly supportive while raising important issues in an objective and clear way. It was well organized and easy to follow. It carefully considered a range of issues, including supplementary data and figure clarity.

Basic reporting

The writing is very clear and the literature review is comprehensive.

Experimental design

This is a very well-written paper that takes advantage of a unique field situation to implement a pre- post design to evaluate the efficacy of Feather Friendly (FF) bird protection materials again ORNILUX bird protection. More intervention studies of bird protection are welcomed. This one has a bit of a challenge in that the FF intervention is compared to another bird-friendly design (ORNILUX) instead of a true untreated control group, pre and post. The choice of ORNILUX for the comparison group was really dictated by the site, which provided a spatially close building to the treatment area, which happened to have ORNILUX installed. Ideally, the authors would have had an untreated control building as well, but that is not how the real world provides buildings. Nevertheless, the efficacy of FF is demonstrated, despite the fact that it was competing against another efficacious treatment. The findings are what one expects to see for a pre-post design with a comparison group. The FF treatment shows fewer collisions than its own pretreated state and the decline in collisions for FF across time is steeper than the change in ORNILUX, which happens to show a slight increase.

Validity of the findings

There are a few concerning features that I would encourage the authors to address. First, the fact that the authors collapsed the systematic with the incidental observations could be problematic if the incidental observations were biased in favor of greater detection of carcasses for the pre-treatment areas. That is, if everyone was concerned about the amount of the bird death at the Science Complex and they really paid attention to carcasses between systematic observation days there, but not at the comparison building, then results could be biased by the inclusion of the incidental reports. The only way to really tell is to redo the analyses without the incidental reports. I suspect the pre- post significant difference at the Science complex would be sustained (the change would be 34 pre-treatment to 2 post-treatment). But I’m not sure the numbers would be sufficient for model convergence for the comparison site, which changed from 3 to 4 collisions across the two periods. The reason one adopts a systematic observation schedule is to avoid these kinds of challenges to claims of internal validity. The fact that incidental reports were treated as if they were systematic, and that there were so many of them, despite the fairly frequent systematic observation time interval, was concerning. Anything the authors can provide to assure the reader that this potential bias did not alter the results would be welcomed. Had those incidentals been left on the ground, most would have been detected at the next systematic observation time, I suspect, given how persistent carcasses were when left out for scavengers.
Second, it is not clear how the authors included “feather smears on windows” (line 188) as data, or if they just used these smears as motivations to look especially carefully for carcass evidence. If the smears were in fact used as data, it would be useful minimally to indicate how often that happened. In addition, it would be useful to state how you kept smears from being repeatedly counted. Many window collision studies do not include smears as data. I have not used it because the smears often cannot be removed after observation, because the windows are too high. The concern would be that it can be difficult to note the location of smears to assure that observers only count the smear the first time it is observed. Please comment about whether this could be a problem in this study. I wondered if counting smears as data also explained the unusually large number of unknown species (37 I believe). This may be useful to comment on as well.
Third, I found it difficult to match up what was described in the data file compared to the text. The text mentioned 364 surveys during pre-treatment and 362 post-treatment. I assumed this would mean researchers observed building 1 and 2 364 times during pre-treatment, but this is clearly not the case. There are 1,451 lines of data instead of the larger number that would be required by my reading (i.e. 13 facades X 364 observations = 4,732 lines of data in the Excel file, just for the pre-test, without including separate lines of incidental observations). It would help the reader if you can clarify several things here, in case other readers want to replicate your findings. Please clarify whether, during systematic observation, all facades were observed once and each observation led to an entry in the Excel file. I notice that "building.side" is not evenly distributed, so there are clearly not an equal number of observations across facades. The data show 693 occurrences of something labeled b1.1, which I believe should be building 1, façade 1. But then there is only 1 for b1.10? and 2 for b1.11? In general, the excel file needs more information to help the reader identify the variables. I would recommend simply adding a tab where you define each variable. Also, it would help to create a text box that contains the commands that were actually run for the models tested. I am assuming, for example, that your dependent or outcome variable is "count.area", based on what I read in the text, but it would be good to see the actual commands to confirm this guess.
One not-so-common innovation in the study was useful, and I would encourage the authors to bring more attention to it. The authors used façade as their unit of analysis, not building. This was a smart choice, in that facades on the same building often radically differ in their collision numbers, so it makes sense to focus on facades to understand variability. This could make targeting of treatment areas more efficient and cost-effective. It also supplies the data analyst with more “level 2” units (using HLM terms) within studies that contain small-scale interventions. This can help with model convergence in small studies. I wish I had realized this opportunity when conducting my own studies, but I was influence by what was standard practice back then of treating each building separately. The authors noted that façade 8 had a high number of collisions, but it would be useful to give a sense of the numbers of collisions across other facades as well.

Additional comments

The rest of my comments are minor matters that can be addressed with word changes, sentence editing, or minor additions.

Minor points (identified by line number)
52, 56, and 61: There is a tendency to rely on the term “urbanization” to define risky areas for birds. I would encourage use of the term “developed” instead. Otherwise, readers might think the authors believe only urban centers pose risks. Some spectacular risks, like at the world trade center, have been linked to urban areas, but other research shows how risks are especially high for one- and two-story buildings, which can be found in abundance in suburban and rural areas. In fact, some have found rural areas more risky than urban (Kummer et al. 2016) and that large buildings in low-urbanized areas may be especially risky (Hager et al. 2017). No building with windows is off the hook for responsibility for bird collisions, so it is useful to use words that blame buildings, not necessarily the urban location of the building. If the authors have data suggesting we should assign more responsibility to urban locations, then please share it.

107 and elsewhere: ORNILUX is described as not visible to humans, when the application is visible if one gets close to an ORNILUX window. Or, if the building used a truly invisible ORNILUX fabrication, it would be useful to describe the particular name or model number that is invisible.

135: I recommend ending the sentence with: “, based on tunnel testing.” This helps underscore your point that current recommendations for bird-friendly glass treatments are seldom tested in the field.

172: As noted above, it was difficult to figure out how many observations of each façade of the building occurred. It would be useful if you could say “all study facades of both buildings” were systematically observed on the same day, between 2 and 5 times per week” or something to that effect.
162: I recommend here or elsewhere that you report on the cost and funder for the intervention. Lots of people who read bird collisions studies ask me how to get these interventions funded and I have no good answer, except to encourage other researchers to include these details.
207: Thanks for including all your software choices. For clarity, I recommend pulling out the one that did the heavy lifting on the mixed model. Something like: “The generalized linear mixed model utilized the lme4 package in R”
210: Never heard of a bobyqa optimizer, but it sounds like something that might be useful in small studies? Can you please supply a few words of description for it?
262: This sentence could be clearer. Perhaps you could add but not “compared with treated glass in” 2016-2018
288: I’m not sure if it is generally thought that ORNILUX or fritted glass would need exterior protection in addition to what they already have, if this is what the author is suggesting. For example, fritted glass has the protection baked into the glass, not on the exterior surface, and it is still effective. So I am not sure that every preventive measure is now thought to need exterior markings as well. It is an interesting question, so perhaps it could be brought up as an area in need of future research. What trade-offs exist for the less aesthetically pleasing fritted glass vs. markers on the exterior of the glass. Which would protect better? It is not known and would be difficult to know in a field setting with pre- and post-test.
293: Can you please provide a cite for commercially available UV markers for glass exteriors? I know that American Bird Conservancy has listed some of the stick-on markers as effective, but it was not clear how long such temporary stickers last and how effective they are (the ABC web site does not list a threat rating for them).
298: Again, unless the version of ORNILUX you are working with is different from the one I have seen, it would be more accurate to say they offer “relatively unobstructed views.”
301: Another limitation might be cost. I was told that adding UV windows would add 1% to construction costs over standard windows, which is pretty costly. I gather that the authors do not think retrofits are costly though, so maybe newer materials are cheaper. It would be good to supply more cost details, if possible.
334: time-consuming
348: a citation is needed here—you could use the ABC site for model legislation or building standards.
353: cost-effective? My experience is very different from what is suggested here. Only if the death of a bird can be assigned a high cost would this a fritted retrofit be considered cost-effective where I work. A retrofit implies that clear glass windows were installed but due to bird deaths are replaced with fritted windows. First, people object to fritted glass creating a blurry view; if they have had a clear view that gets removed and frits installed, then the difference is really noticeable. I imagine there would be lots of worker complaints about this solution, even if it saves birds and energy costs. The newly constructed building I examined with fritted windows was relatively safe for birds but the frits produced lots of anecdotal complaints from building users. Second, I was told that no windows ever would be replaced on any building unless the building itself would be renovated, in order to save costs and fit within the building renovation schedule. If you have a more hopeful story about how the windows received frits at this site and details of the funding amount and source for the Feather Friendly intervention, it would be useful to hear. When I was trying to fund an intervention I was cautioned that many environmental donations are less than $10,000, which means that many office buildings would be too costly to mitigate, unless fundraising is unusually good. I would love to hear about why the authors think this is relatively easy to accomplish. Perhaps the lawsuit in Canada makes building owners take bird deaths more seriously?
362: Again, it would be useful to flesh out the details with examples of what product would be available to protect residential windows for less than $40.00.
389 and elsewhere (e.g., 429, 457, 467, 470, 473, 489): Citations should not have capitalized words across the title.
442 something is wrong with the repeated “preprints” in this reference. Should not be a preprint from this date but rather a full published citation.
512 Connell should be O’Connell.
Table 1: The text needs a closed parenthesis before the semicolon: )); It would be useful to include what the outcome variable is. I believe it is not the raw number of collisions but rather collisions by glass area, yes?
Figure 1: dikes, not dykes. Generally, I find it useful in scientific papers to avoid identifying things by color, given that readers may print out papers on black and white. This would be time-consuming here so I’m just leaving it as a suggestion.
Figure 4: Requires a color print to match the wording in the text; ideally, this could be conveyed with some other graphic, such as hashed lines or dots. I am not used to seeing non-linear graphs presenting the same spacing between .01 and .10 as between .10 and 1.0 along the bottom axis. Is this based on a log-transform? Could you explain that in the text that accompanies the figure?
I wonder if it would be easier to get your point across by replacing the Figure with a table that would present the raw numbers of collisions and the IRRs (95% CI). For example, you could have one row for FF and one for ORNILUX. The columns would contain the pre-treatment collision numbers, post-treatment collision numbers, and IRR values (95% CI). In my minds eye, such a graphic would get across the strong results in a quicker fashion, instead of having the reader wonder about the metric along the bottom axis. Just a suggestion; this may be the preferred style of presenting IRR data from this particular statistical package, which I do not use.

References

Hager SB, Cosentino BJ, Aguilar-Gómez MA, Anderson ML, Bakermans M, Boves TJ, Brandes D, Butler MW, Butler EM, and Cagle NL. 2017. Continent-wide analysis of how urbanization affects bird-window collision mortality in North America. Biological Conservation 212:209-215. DOI 10.1016/j.biocon.2017.06.014
Kummer JA, Bayne EM, and Machtans CS. 2016. Use of citizen science to identify factors affecting bird–window collision risk at houses. The Condor 118:624-639. 10.1650/CONDOR-16-26.1

·

Basic reporting

The article is well-written and clear. I have made some specific comments on how to improve it below.

Experimental design

The experimental design is appropriate and follows standards form other bird-window collision studies. For the statistical analyses, I suggest adding an additional variable, please see specific comments below.

Validity of the findings

These findings are much needed and give us hope that we can reduce collisions with the appropriate collision deterrents. The findings are sound and result from disciplined sampling and data analyses.

Additional comments

You have a really nice paper showing some great results from a longish-term dataset, cool stuff! I like how you have kept it simple and clear, and you made a real effort to report all the previous literature, excellent!

I have two major comments and revisions I would like for you to make, which I believe will make your paper stronger and even more clear to the readers:

1. Incorporate % glass into your models. Calculate the % glass cover for each of your buildings and use that as a variable in your GLMMs. This will help you control for the differences in proportion of glass in each of the buildings, and it will help understand how this variable interacts with others. This is important because % glass has been proven to affect collisions, besides just total glass area.
2. Re-structure and re-write discussion. Currently, your discussion and conclusions have all that is needed, but the text is rather disorganized and way too long in conclusions. In the discussion section I made a specific suggestion as to how you might structure your discussion.

One last general comment: I still have one minor confusion which is why you divided the Ornilux data in 2013-2015 and 2016-2018. Maybe you need to do a better job at explaining why these data were separated in periods because there was no before-after for this building. My understanding is that you compared the Ornilux 2013-2015 to the untreated Science building. But then what did you compare Ornilux 2016-2018 with? Perhaps this is already in the paper but it was not so clear to me, which might be similar to the readers.


Introduction

You have done a really good job at summarizing and reporting the existing literature and knowledge on bird-window collisions. This introduction is well-written, appropriately supported by literature, and adequate for your paper, well done!

Ln 95 – Maybe also mention the UV-reflective films that are available
Ln 103 – Is there a citation that could back this up? I don’t personally know of one but there might be one out there…
Ln 122 – Please also review https://peerj.com/articles/4215/
Ln 137 – Perhaps a single main objective here would be good such as “Evaluate the effectiveness of glass treatments to prevent bird-window collisions”, and then you can describe the two products and situations (before-after, only-after) that you studied.

Methods


Statistical analyses are sound and appropriate for the questions you are asking. But I suggest you add % glass as a variable to the models and re-run everything.

Ln 159 – It’d be great to have the % glass which you can get by calculating the façade area and using the glass area you have already calculated
Ln 162 (Figure 2C) – Specify which panel of the figure shows the specific thing you are mentioning, in this case the retrofit
Ln 171 – Same as above. Calculate % glass
Ln 175 – Before you say 730 surveys. Please check the numbers throughout

Results
Ln 230-240 – All this text can be summarized in a table that shows collisions per building, time period, and has columns for survey or incidental. Please provide this summary table and in the text only highlight important findings, for example the % collision reduction after retrofit.


Discussion

You have most of the important points between the discussion and the conclusion text, but it is rather disorganized currently. I suggest you write an outline of the discussion (subtitles which you could delete later) and start from local to global. So, start by talking about your results and how they relate to other studies (which you have done). Then talk about collision deterrents in general and its limitations (cost, aesthetics), and maybe close with the part about policy, public engagement, and some of that text you have in the conclusions

Ln 316 – Exactly…so you need to report this difference and maybe account for it in your models

Conclusions

Here you need to start strong and report that FF reduced collisions by 92% in a before and after study, while Ornilux only in xx% compared to an untreated similar building. That should be the very first sentence.

Your conclusions are way too long and read more like a discussion. Move some of this text to the discussion (only if really required there) and leave here just the results of your study, and perhaps a recommendation sentence for future studies. Keep it short and clear. Should not be longer than 4 short sentences. Conclusions are better off without references, keep these in discussion.

---

## Round 0.2 · Minor Revisions

Only one reviewer was available to examine the changes in your manuscript. This reviewer and I agree that the changes have improved an already strong manuscript. The reviewer has several suggestions for clearer wording, and I have noted a few minor grammatical corrections in the attached pdf.

Reviewer 1 ·

Basic reporting

In general, this is a much clearer version of the manuscript. It is especially reassuring to know that systematic and anecdotal observations were not mixed together in the analyses. The ms. was strong to begin with and a few more revisions should make it a bit more reader-friendly.

Experimental design

The design is fine.

Validity of the findings

Findings are valid but I have some suggested rewording. I only have one area of major concern and apologize that I did not raise this earlier. I believe it is just a clarity issue, which can be addressed with minor editing. Below I have pulled original text from the ms. and Table 1 and provided a suggested set of revisions, with my comments to explain why the revisions were needed, in my view.

Original, Line 229 -231
We evaluated support for three models as shown in Table 1. Using a base model with random-
effects (year and observer) and season, we compared a building-only effect model, season and a
treatment effects model.
Original:
Table 1
Fixed effects: treat.grp = building by year treatment Science Complex Pre-FF (2013-2015), Science Complex Post-FF (2016-2018), Annex Ornilux (2013-2015), Annex Ornilux (2016-2018); building = Science Complex or Annex; season=Winter, Fall, Spring or Summer.
Random effects: 1|year = monitoring year, 1|obs = observer. Offset = window area ΔAIC =change in AIC relative to top model, w = Akaike weight., and df = degrees freedom.

Model structure AIC ΔAIC w df
treat.grp + season + (1|year) + (1|obs) 335.95 0 1.00 9
season + (1|year) + (1|obs) 349.63 13.68 0 6
building + season + (1|year) + (1|obs) 350.25 14.30 0 7
(1|year) + (1|obs) 352.78 16.83 0 3

Suggested revision, line 229
We compared a baseline model, which contained only year and observer random effects, against three alternatives that added fixed effects. These additions included tests of building and season, season only, and the full model with treatment effects, which was represented by the combination of building and time period.
Suggested revision, Table 1: I added the “Row” column to Table 1 to ease referencing.
Row Model structure AIC ΔAIC w df
A treat.grp + season + (1|year) + (1|obs) 335.95 0 1.00 9
B season + (1|year) + (1|obs) 349.63 13.68 0 6
C building + season + (1|year) + (1|obs) 350.25 14.30 0 7
D (1|year) + (1|obs) 352.78 16.83 0 3

Comments
I suggest the revisions above because it seems odd to see that Table 1 has 4 rows of effects but line 229 refers to 3 models. Also, the sentence, “Using a base model with random-
effects (year and observer) and season, we compared a building-only effect model, season and a
treatment effects model” does not seem accurate, because the base model is row D, the random effects only, yes? Here the AIC is largest, so you are progressively accounting for more variability by adding the fixed effects for rows C, then B, then A.

Original, Lines 255-258
When competing models based on standardized survey data corrected for differences in area of
glass were compared, the best supported model for predicting collision risk was the model that
included the building's treatment group, versus the building-only or base model (year, observer
and season; Table 1).

Suggested revision, Line 255:
Competing models, based on standardized survey data corrected for differences in area of
glass, compared simpler models to the full model. The full model, testing the combination of building and time period, demonstrated the best model fit. The full effects model provided the greatest reduction in AIC, compared to a random effects model testing year and observer only (Row D, Table 1), a model adding season alone (Row B), and a model adding building and season (Row C).

Comments
I suggested this revision because it was confusing for me to try to figure out why something called a “building-only or base model” would have a fixed effect (building) in the base model, given that you have the more basic Row D. Also, as I read the Rows, there is not a Building-only model, which I assume would have building, year, and observer. It was not clear to me whether your phrase “building-only or base model” was referring to one or two models (is the building-only the base model? Or are you describing a building only model in addition to a base model?).

Also, you note that I suggest you refer to the treatment effect as the combination of building by period instead of building by year (“treat.grp = building by year”), because your data (although using words, not numbers) and degrees of freedom look like you handled the three pre-treatment years and the three post-treatment years as two different periods. That is, with numeric coding your period variable would be coded 0, 1 (for 2013-2015 and 2016-2018) not 1-6 for years 2013-2018.

I do not care that you adopt these exact wording changes, but I hope that, by bringing up these suggested changes, you can see where the reader might be confused when trying to match the table to the text. If I were writing this I would refer to a “building by period interaction,” but I understand that phrasing derives from an old analysis of variance framework, which you did not explicitly test (i.e. by including a period-only main effect model). So I referred instead to a combination of building and time period.

Additional comments

Minor points/suggested wording changes for clarity (by line number from revised submission)

89-90. I’d omit “that they fail to perceive and avoid.” You make the full point in the next sentence, which is very clear. This sentence makes me think that birds are failing to perceive an unobstructed route, when in fact they are falsely perceiving an unobstructed route.

102: by “static object” I think you are referring to the markers but this could be clearer. I suggest: ,and therefore the color of the visual markers must provide …

218: Data were, not data was

286: “recommendations that were” change to “these recommendations were”

350: American English would be “fueled,” not “fueled.” Not sure which one PeerJ wants.

364: change ; to , or perhaps make text past the semicolon into a new sentence, given how long this sentence is.

381: change are to is

---

## Round 0.3 · accepted · Accept

Thanks for your rapid response to the reviewer suggestions. I consider the manuscript now ready for publication.